# Effective Use of Variational Embedding Capacity in Expressive End-to-End Speech Synthesis

## Abstract

Recent work has explored sequence-to-sequence latent variable models for expressive speech synthesis (supporting control and transfer of prosody and style), but has not presented a coherent framework for understanding the trade-offs between the competing methods. In this paper, we propose embedding capacity (the amount of information the embedding contains about the data) as a unified method of analyzing the behavior of latent variable models of speech, comparing existing heuristic (non-variational) methods to variational methods that are able to explicitly constrain capacity using an upper bound on representational mutual information. In our proposed model (Capacitron), we show that by adding conditional dependencies to the variational posterior such that it matches the form of the true posterior, the same model can be used for high-precision prosody transfer, text-agnostic style transfer, and generation of natural-sounding prior samples. For multi-speaker models, Capacitron is able to preserve target speaker identity during inter-speaker prosody transfer and when drawing samples from the latent prior. Lastly, we introduce a method for decomposing embedding capacity hierarchically across two sets of latents, allowing a portion of the latent variability to be specified and the remaining variability sampled from a learned prior. Audio examples are available on the web[1].

## 1 Introduction

The synthesis of realistic human speech is a challenging problem that is important for natural human-computer interaction. End-to-end neural network-based approaches have seen significant progress in recent years (Wang et al., 2017; Taigman et al., 2018; Ping et al., 2018; Sotelo et al., 2017), even matching human performance for short assistant-like utterances (Shen et al., 2018). However, these neural models are sometimes viewed as less interpretable or controllable than more traditional models composed of multiple stages of processing that each operate on reified linguistic or phonetic representations.

Text-to-speech (TTS) is an underdetermined problem, meaning the same text input has an infinite number of reasonable spoken realizations. In addition to speaker and channel characteristics, important sources of variability in TTS include intonation, stress, and rhythm (collectively referred to as *prosody*). These attributes convey linguistic, semantic, and emotional meaning beyond what is present in the lexical representation (i.e., the text) (Wagner & Watson, 2010). Recent end-to-end TTS research has aimed to model and/or directly control the remaining variability in the output.

Skerry-Ryan et al. (2018) augment a Tacotron-like model (Wang et al., 2017) with a deterministic encoder that projects reference speech into a learned embedding space. The system can be used for prosody transfer between speakers ("say it like this"), but does not work for transfer between unrelated sentences, and does not preserve the pitch range of the target speaker. Lee & Kim (2019) partially address the pitch range problem by centering the learned embeddings using speaker-wise means.

---

[1]`https://variational-embedding-capacity.github.io/demos/`

Other work targets *style* transfer, a text-agnostic variation on prosody transfer. The Global Style Token (GST) system (Wang et al., 2018) uses a modified attention-based reference encoder to transfer global style properties to arbitrary text, and Ma et al. (2019) use an adversarial objective to disentangle style from text.

Hsu et al. (2019) and Zhang et al. (2019) use a variational approach (Kingma & Welling, 2014) to tackle the style task. Advantages of this approach include its ability to generate style samples via the accompanying prior and the potential for better disentangling between latent style factors (Burgess et al., 2018). Additionally, Hsu et al. (2019) use a Gaussian mixture prior over the latents, which (when interpreting the mixture component index as a high-level discrete latent) allows a form of hierarchical control.

This work extends the above approaches by providing the following contributions:

1. We propose a unified approach for analyzing the characteristics of TTS latent variable models, independent of architecture, using the *capacity* of the learned embeddings (i.e., the representational mutual information between the embedding and the data).

2. We target specific capacities for our proposed model using a Lagrange multiplier-based optimization scheme, and show that capacity is correlated with perceptual reference similarity.

3. We show that modifying the variational posterior to match the form of the true posterior enables style and prosody transfer in the same model, helps preserve target speaker identity during inter-speaker transfer, and leads to natural-sounding prior samples even at high embedding capacities.

4. We introduce a method for controlling what fraction of the variation represented in an embedding is specified, allowing the remaining variation to be sampled from the model.

## 2 MEASURING REFERENCE EMBEDDING CAPACITY

### 2.1 LEARNING A REFERENCE EMBEDDING SPACE

Existing heuristic (non-variational) end-to-end approaches to prosody and style transfer (Skerry-Ryan et al., 2018; Wang et al., 2018; Lee & Kim, 2019; Henter et al., 2018) typically start with the teacher-forced reconstruction loss, (1), used to train Tacotron-like sequence-to-sequence models and simply augment the model with a deterministic reference encoder, $g_e(\mathbf{x})$, as shown in eq. (2).

$$L(\mathbf{x}, \mathbf{y}_\mathrm{T}, \mathbf{y}_\mathrm{S}) \equiv -\log p(\mathbf{x}|\mathbf{y}_\mathrm{T}, \mathbf{y}_\mathrm{S}) = \|f_\theta(\mathbf{y}_\mathrm{T}, \mathbf{y}_\mathrm{S}) - \mathbf{x}\|_1 + K \tag{1}$$

$$L'(\mathbf{x}, \mathbf{y}_\mathrm{T}, \mathbf{y}_\mathrm{S}) \equiv -\log p(\mathbf{x}|\mathbf{y}_\mathrm{T}, \mathbf{y}_\mathrm{S}, g_e(\mathbf{x})) = \|f_\theta(\mathbf{y}_\mathrm{T}, \mathbf{y}_\mathrm{S}, g_e(\mathbf{x})) - \mathbf{x}\|_1 + K \tag{2}$$

where $\mathbf{x}$ is an audio spectrogram, $\mathbf{y}_\mathrm{T}$ is the input text, $\mathbf{y}_\mathrm{S}$ is the target speaker (if training a multi-speaker model), $f_\theta(\cdot)$ is a deterministic function that maps the inputs to spectrogram predictions, and $K$ is a normalization constant. Teacher-forcing implies that $f_\theta(\cdot)$ is dependent on $\mathbf{x}_{<t}$ when predicting spectrogram frame $\mathbf{x}_t$. In practice, $f_\theta(\cdot)$ serves as the greedy deterministic output of the model, and transfer is accomplished by pairing the embedding computed by the reference encoder with different text or speakers during synthesis.

In these heuristic models, the architecture chosen for the reference encoder determines the transfer characteristics of the model. This decision affects the information capacity of the embedding and allows the model to target a specific trade-off between transfer *precision* (how closely the output resembles the reference) and *generality* (how well an embedding works when paired with arbitrary text). Higher capacity embeddings prioritize precision and are better suited for prosody transfer to similar text, while lower capacity embeddings prioritize generality and are better suited for text-agnostic style transfer.

The variational extensions from Hsu et al. (2019) and Zhang et al. (2019) augment the reconstruction loss in eq. (2) with a KL divergence term. This encourages a stochastic reference encoder (variational posterior), $q(\mathbf{z}|\mathbf{x})$, to align well with a prior, $p(\mathbf{z})$ (eq. (3)). The overall loss is then equivalent to the negative evidence lower bound (ELBO) of the marginal likelihood of the data (Kingma & Welling, 2014).

$$L_\mathrm{ELBO}(\mathbf{x}, \mathbf{y}_\mathrm{T}, \mathbf{y}_\mathrm{S}) \equiv E_{\mathbf{z} \sim q(\mathbf{z}|\mathbf{x})}[-\log p(\mathbf{x}|\mathbf{z}, \mathbf{y}_\mathrm{T}, \mathbf{y}_\mathrm{S})] + D_\mathrm{KL}(q(\mathbf{z}|\mathbf{x})\|p(\mathbf{z})) \tag{3}$$

$$-\log p(\mathbf{x}|\mathbf{y}_\mathrm{T}, \mathbf{y}_\mathrm{S}) \le L_\mathrm{ELBO}(\mathbf{x}, \mathbf{y}_\mathrm{T}, \mathbf{y}_\mathrm{S}) \tag{4}$$

Controlling embedding capacity in variational models can be accomplished more directly by manipulating the KL term in (3). Recent work has shown that the KL term provides an upper bound on the mutual information between the data, $\mathbf{x}$, and the latent embedding, $\mathbf{z} \sim q(\mathbf{z}|\mathbf{x})$ (Hoffman & Johnson, 2016; Makhzani et al., 2015; Alemi et al., 2018).

$$R^{\text{AVG}} \equiv E_{\mathbf{x} \sim p_D(\mathbf{x})}[D_{\text{KL}}(q(\mathbf{z}|\mathbf{x})\|p(\mathbf{z}))], \qquad R \equiv D_{\text{KL}}(q(\mathbf{z}|\mathbf{x})\|p(\mathbf{z})) \qquad (5)$$

$$I_q(\mathbf{X};\mathbf{Z}) \equiv E_{\mathbf{x} \sim p_D(\mathbf{x})}[D_{\text{KL}}(q(\mathbf{z}|\mathbf{x})\|q(\mathbf{z}))], \qquad q(\mathbf{z}) \equiv E_{\mathbf{x} \sim p_D(\mathbf{x})}q(\mathbf{z}|\mathbf{x}) \qquad (6)$$

$$R^{\text{AVG}} = I_q(\mathbf{X};\mathbf{Z}) + D_{\text{KL}}(q(\mathbf{z})\|p(\mathbf{z})) \qquad (7)$$

$$\implies I_q(\mathbf{X};\mathbf{Z}) \leq R^{\text{AVG}} \qquad (8)$$

where $p_D(\mathbf{x})$ is the data distribution, $R$ is the the KL term in (3), $R^{\text{AVG}}$ is the KL term averaged over the data distribution, $I_q(\mathbf{X};\mathbf{Z})$ is the representational mutual information (the *capacity* of $\mathbf{z}$), and $q(\mathbf{z})$ is the *aggregated posterior*. This brief derivation is expanded in Appendix C.1.

The bound in (8) follows from (7) and the non-negativity of the KL divergence, and (7) shows that the slack on the bound is $D_{\text{KL}}(q(\mathbf{z})\|p(\mathbf{z}))$, the *aggregate KL*. In addition to providing a tighter bound, having a low aggregate KL is desirable when sampling from the model via the prior, because then the samples of $\mathbf{z}$ that the decoder sees during training will be very similar to samples from the prior.

Various approaches to controlling the KL term have been proposed, including varying a weight on the KL term, $\beta$ (Higgins et al., 2017), and penalizing its deviation from a target value (Alemi et al., 2018; Burgess et al., 2018). Because we would like to smoothly optimize for a specific bound on the embedding capacity, we adapt the Lagrange multiplier-based optimization approach of Rezende & Viola (2018) by applying it to the KL term rather than the reconstruction term.

$$\min_\theta \max_{\beta \geq 0} \left\{ E_{\mathbf{z} \sim q_\theta(\mathbf{z}|\mathbf{x})}[-\log p_\theta(\mathbf{x}|\mathbf{z}, \mathbf{y}_\text{T}, \mathbf{y}_\text{S})] + \beta(D_{\text{KL}}(q_\theta(\mathbf{z}|\mathbf{x})\|p(\mathbf{z})) - C) \right\} \qquad (9)$$

where $\theta$ are the model parameters, $\beta$ serves as an automatically-tuned weight on the KL term, $C$ is the capacity limit, and updates to $\theta$ and $\beta$ are interleaved during training. We constrain $\beta$ to be non-negative by passing an unconstrained parameter through a softplus non-linearity, which makes the capacity constraint a limit rather than a target. This approach is less tedious than tuning $\beta$ by hand and leads to more consistent behavior from run-to-run. It also allows more stable optimization than directly penalizing the $\ell_1$ deviation from the target KL.

## 2.2 ESTIMATING EMBEDDING CAPACITY

**Estimating heuristic embedding capacity** Unfortunately, the heuristic methods do not come packaged with an easy way to estimate embedding capacity. We can estimate an effective capacity *ordering*, however, by measuring the test-time reconstruction loss when using the reference encoder from each method. In Figure 1, we show how the reconstruction loss varies with embedding dimensionality for the tanh-based prosody transfer (PT) and softmax-based global style token (GST) bottlenecks (Skerry-Ryan et al., 2018; Wang et al., 2018) and for variational models (Var.) with different capacity limits, $C$. We also compare to a baseline Tacotron model without a reference encoder. For this preliminary comparison, we use the expressive single-speaker dataset and training setup described in Section 4.2. Looking at the heuristic methods in Figure 1, we see that the GST bottleneck is much more restrictive than the PT bottleneck, which hurts transfer precision but allows sufficient embedding generality for text-agnostic style transfer.

**Bounding variational embedding capacity** We saw in (8) that the KL term is an upper bound on embedding capacity, so we can directly target a specific capacity limit by constraining the KL term using the objective in eq. (9). For the three values of $C$ in Figure 1, we can see that the reconstruction loss flattens out once the embedding reaches a certain dimensionality. This gives us a consistent way to control embedding capacity as it only requires using a reference encoder architecture with sufficient structural capacity (at least $C$) to achieve the desired representational capacity in the variational embedding. Because of this, we use 128-dimensional embeddings in all of our experiments, which should be sufficient for the range of capacities we target.

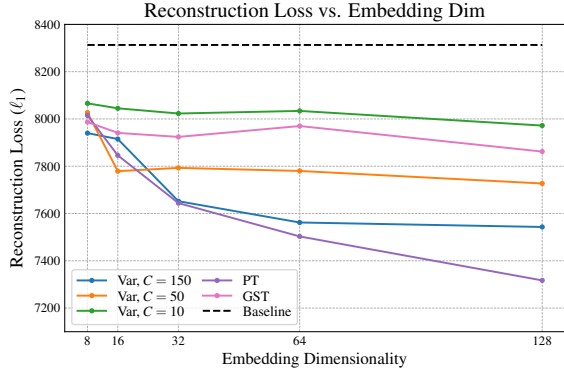

Figure 1: Reconstruction loss vs. embedding dimensionality for a variety of heuristic and variational models. For the variational model (Var.), we vary the capacity limit, $C$. Notice how the reconstruction loss flattens out at lower values for higher values of $C$. The heuristic models are denoted by PT and GST for Prosody Transfer and Global Style Tokens. Figure B.1 in the appendix shows how the KL term changes when varying $C$ as well as the KL weight, $\beta$.

## 3 MAKING EFFECTIVE USE OF EMBEDDING CAPACITY

### 3.1 MATCHING THE FORM OF THE TRUE POSTERIOR

In previous work (Hsu et al., 2019; Zhang et al., 2019), the variational posterior has the form $q(\mathbf{z}|\mathbf{x})$, which matches the form of the true posterior for a simple generative model $p(\mathbf{x}|\mathbf{z})p(\mathbf{z})$. However, for the conditional generative model used in TTS, $p(\mathbf{x}|\mathbf{z}, \mathbf{y}_\mathrm{T}, \mathbf{y}_\mathrm{S})p(\mathbf{z})$, it is missing conditional dependencies present in the true posterior, $p(\mathbf{z}|\mathbf{x}, \mathbf{y}_\mathrm{T}, \mathbf{y}_\mathrm{S})$. Figure 2 shows this visually. In order to

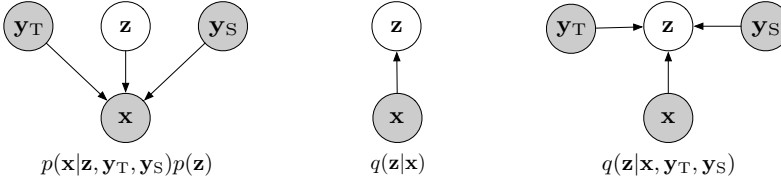

Figure 2: Adding conditional dependencies to the variational posterior. Shaded nodes indicate observed variabes. [left] The true generative model. [center] Variational posterior missing conditional dependencies present in the true posterior. [right] Variational posterior that matches the form of the true posterior.

match the form of the true posterior, we inject information about the text and the speaker into the network that predicts the parameters of the variational posterior. Speaker information is represented as learned speaker-wise embedding vectors, while the text information is summarized into a vector by passing the output of the Tacotron text encoder through a unidirectional RNN as done by Stanton et al. (2018). Appendix A.1 gives additional details.

For this work, we use a simple diagonal Gaussian for the approximate posterior, $q(\mathbf{z}|\mathbf{x}, \mathbf{y}_\mathrm{T}, \mathbf{y}_\mathrm{S})$ and a standard normal distribution for the prior, $p(\mathbf{z})$. We use these distributions for simplicity and efficiency, but using more powerful distributions such as Gaussian mixtures or normalizing flows (Rezende & Mohamed, 2015) should decrease the aggregate KL, leading to better prior samples.

Because we are learning a conditional generative model, $p(\mathbf{x}|\mathbf{y}_\mathrm{T}, \mathbf{y}_\mathrm{S})$, we could have used a learned conditional prior, $p(\mathbf{z}|\mathbf{y}_\mathrm{T}, \mathbf{y}_\mathrm{S})$, in order to improve the quality of the output generated when sampling via the prior. However, in this work we focus on the transfer use case where we infer $\mathbf{z}^\mathrm{ref} \sim q(\mathbf{z}|\mathbf{x}^\mathrm{ref}, \mathbf{y}_\mathrm{T}^\mathrm{ref}, \mathbf{y}_\mathrm{S}^\mathrm{ref})$ from a reference utterance and use it to re-synthesize speech using different text or speaker inputs, $\mathbf{x}' \sim p(\mathbf{x}|\mathbf{z}^\mathrm{ref}, \mathbf{y}_\mathrm{T}', \mathbf{y}_\mathrm{S}')$. Using a fixed prior allows $\mathbf{z}$ to share a high probability region across all text and speakers so that an embedding inferred from one utterance is likely to lead to non-degenerate output when being used with any other text or speaker.

## 3.2 DECOMPOSING EMBEDDING CAPACITY HIERARCHICALLY

In inter-text style transfer uses cases, we infer $\mathbf{z}^{\text{ref}}$ from a reference utterance and then use it to generate a new utterance with the same style but different text. One problem with this approach is that $\mathbf{z}^{\text{ref}}$ completely specifies all variation that the latent embedding is capable of conveying to the decoder, $p(\mathbf{x}|\mathbf{z}^{\text{ref}}, \mathbf{y}_{\text{T}}, \mathbf{y}_{\text{S}})$. So, even though there are many possible realizations of an utterance with a given style, this approach can produce only one[2].

To address this issue, we decompose the latents, $\mathbf{z}$, hierarchically (Sønderby et al., 2016) into high-level latents, $\mathbf{z}_{\text{H}}$, and low-level latents, $\mathbf{z}_{\text{L}}$, as shown in Figure 3. This differs from the hierarchical interpretation of the Gaussian mixture prior used by Hsu et al. (2019) in that here the high-level latents are continuous vectors rather than a single categorical variable. Factorizing continuous latents in this way allows us to specify how the joint capacity, $I_q(\mathbf{X}; [\mathbf{Z}_{\text{H}}, \mathbf{Z}_{\text{L}}])$, is divided between $\mathbf{z}_{\text{H}}$ and $\mathbf{z}_{\text{L}}$. This approach can also be extended to additional levels of latents, each containing a prescribed proportion of the overall joint capacity.

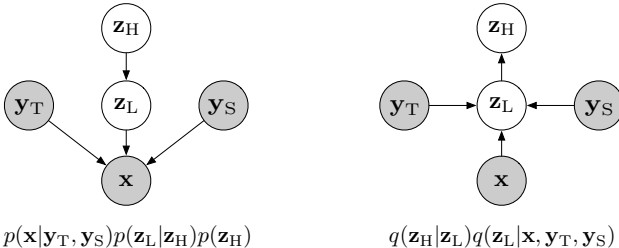

$$p(\mathbf{x}|\mathbf{y}_{\text{T}}, \mathbf{y}_{\text{S}})p(\mathbf{z}_{\text{L}}|\mathbf{z}_{\text{H}})p(\mathbf{z}_{\text{H}}) \qquad\qquad q(\mathbf{z}_{\text{H}}|\mathbf{z}_{\text{L}})q(\mathbf{z}_{\text{L}}|\mathbf{x}, \mathbf{y}_{\text{T}}, \mathbf{y}_{\text{S}})$$

Figure 3: Hierarchical decomposition of the latents. Shaded nodes indicate observed variables. [left] The true generative model. [right] Variational posterior that matches the form of the true posterior.

As shown in eq. (8), the KL term, $R^{\text{AVG}}$, is an upper bound on $I_q(\mathbf{X}; \mathbf{Z})$. We can also derive similar bounds for $I_q(\mathbf{X}; \mathbf{Z}_{\text{H}})$ and $I_q(\mathbf{X}; \mathbf{Z}_{\text{L}})$. Derivations of these bounds are provided in Appendix C.2.

$$I_q(\mathbf{X}; \mathbf{Z}_{\text{L}}) \leq R^{\text{AVG}} = E_{\mathbf{x} \sim p_D(\mathbf{x})}[D_{KL}(q(\mathbf{z}_{\text{H}}|\mathbf{z}_{\text{L}})q(\mathbf{z}_{\text{L}}|\mathbf{x})\|p(\mathbf{z}_{\text{L}}|\mathbf{z}_{\text{H}})p(\mathbf{z}_{\text{H}}))] \qquad (10)$$

$$I_q(\mathbf{X}; \mathbf{Z}_{\text{H}}) \leq R_{\text{H}}^{\text{AVG}} \equiv E_{\mathbf{x} \sim p_D(\mathbf{x}), \mathbf{z}_{\text{L}} \sim q(\mathbf{z}_{\text{L}}|\mathbf{x})}[D_{KL}(q(\mathbf{z}_{\text{H}}|\mathbf{z}_{\text{L}})\|p(\mathbf{z}_{\text{H}}))] \qquad (11)$$

If we define $R_{\text{L}}^{\text{AVG}} \equiv R^{\text{AVG}} - R_{\text{H}}^{\text{AVG}}$, we end up with the following capacity limits for the hierarchical latents:

$$\implies I_q(\mathbf{X}; \mathbf{Z}_{\text{H}}) \leq R_{\text{H}}^{\text{AVG}}, \qquad I_q(\mathbf{X}; \mathbf{Z}_{\text{L}}) \leq R_{\text{H}}^{\text{AVG}} + R_{\text{L}}^{\text{AVG}} \qquad (12)$$

The negative ELBO for this model can be written as:

$$L_{\text{ELBO}}(\mathbf{x}, \mathbf{y}_{\text{T}}, \mathbf{y}_{\text{S}}) = -E_{\mathbf{z}_{\text{L}} \sim q(\mathbf{z}_{\text{L}}|\mathbf{x})}[\log p(\mathbf{x}|\mathbf{z}_{\text{L}}, \mathbf{y}_{\text{T}}, \mathbf{y}_{\text{S}})] + R_{\text{H}} + R_{\text{L}} \qquad (13)$$

where $R_{\text{H}}$ and $R_{\text{L}}$ are single data point estimates of $R_{\text{H}}^{\text{AVG}}$ and $R_{\text{L}}^{\text{AVG}}$ computed from $\mathbf{x}$. In order to specify how the joint capacity is distributed between the latents, we extend (9) to have two Lagrange multipliers and capacity targets.

$$\min_{\theta} \max_{\beta_{\text{H}}, \beta_{\text{L}} \geq 0} \left\{ E_{\mathbf{z}_{\text{L}} \sim q_\theta(\mathbf{z}_{\text{L}}|\mathbf{x}, \mathbf{y}_{\text{T}}, \mathbf{y}_{\text{S}})}[-\log p_\theta(\mathbf{x}|\mathbf{z}_{\text{L}}, \mathbf{y}_{\text{T}}, \mathbf{y}_{\text{S}})] + \beta_{\text{H}}(R_{\text{H}} - C_{\text{H}}) + \beta_{\text{L}}(R_{\text{L}} - C_{\text{L}}) \right\} \quad (14)$$

$C_{\text{H}}$ limits the information capacity of $\mathbf{z}_{\text{H}}$, and $C_{\text{L}}$ limits how much capacity $\mathbf{z}_{\text{L}}$ has in excess of $\mathbf{z}_{\text{H}}$ (i.e., the total capacity of $\mathbf{z}_{\text{L}}$ is capped at $C_{\text{H}} + C_{\text{L}}$). This allows us to infer $\mathbf{z}_{\text{H}}^{\text{ref}} \sim q(\mathbf{z}_{\text{H}}|\mathbf{z}_{\text{L}})q(\mathbf{z}_{\text{L}}|\mathbf{x}^{\text{ref}}, \mathbf{y}_{\text{T}}^{\text{ref}}, \mathbf{y}_{\text{S}}^{\text{ref}})$ from a reference utterance and use it to sample multiple realizations, $\mathbf{x}' \sim p(\mathbf{x}|\mathbf{z}_{\text{L}}, \mathbf{y}_{\text{T}}, \mathbf{y}_{\text{S}})p(\mathbf{z}_{\text{L}}|\mathbf{z}_{\text{H}}^{\text{ref}})$. Intuitively, the higher $C_{\text{H}}$ is, the more the output will resemble the reference, and the higher $C_{\text{L}}$ is, the more variation we would expect from sample to sample when fixing $\mathbf{z}_{\text{H}}^{\text{ref}}$ and sampling $\mathbf{z}_{\text{L}}$ from $p(\mathbf{z}_{\text{L}}|\mathbf{z}_{\text{H}}^{\text{ref}})$.

---

[2]If the decoder were truly stochastic (not greedy), we could actually sample multiple realizations given the same $\mathbf{z}^{\text{ref}}$, but, at high embedding capacities the variations would likely be very similar perceptually.

## 4 EXPERIMENTS

### 4.1 MODEL ARCHITECTURE AND TRAINING

**Model architecture**  The baseline model we start with is a Tacotron-based system (Wang et al., 2017) that incorporates modifications from Skerry-Ryan et al. (2018), including phoneme inputs instead of characters, GMM attention (Graves, 2013), and a WaveNet neural vocoder (van den Oord et al., 2016) to convert the output mel spectrograms into audio samples (Shen et al., 2018). The decoder RNN uses a reduction factor of 2, meaning that it produces two spectrogram frames per timestep. We use the CBHG text encoder from Wang et al. (2018) and the GMMv2b attention mechanism from Battenberg et al. (2019).

For the heuristic models compared in Section 2.2, we augment the baseline Tacotron with the reference encoders described by Skerry-Ryan et al. (2018) and Wang et al. (2018). For the variational models that we compare in the following experiments, we start with the reference encoder from Skerry-Ryan et al. (2018) and replace the tanh bottleneck layer with an MLP that predicts the parameters of the variational posterior. When used, the additional conditional dependencies (text and speaker) are fed into the MLP as well.

**Training**  To train the models, the primary optimizer is run synchronously across 10 GPU workers (2 of them backup workers) for 300,000 training steps with an effective batch size of 256. It uses the Adam algorithm (Kingma & Ba, 2015) with a learning rate that is annealed from $10^{-3}$ to $5 \times 10^{-5}$ over 200,000 training steps. The optimizer for $\beta$ is run asychronously on the 10 workers and uses SGD with momentum 0.9 and a fixed learning rate of $10^{-5}$. The updates for these two optimizers are interleaved, allowing $\beta$ to converge to a steady state value that achieves the target value for the KL term, as demonstrated in Figure B.2 in the appendix. Additional architectural and training details are provided in Appendix A.

### 4.2 EXPERIMENT SETUP

**Datasets**  For single-speaker models, we use an expressive English language audiobook dataset consisting of 50,086 training utterances (36.5 hours) and 912 test utterances spoken by Catherine Byers, the speaker from the 2013 Blizzard Challenge. Multi-speaker models are trained using high-quality English data from 58 voice assistant-like speakers, consisting of 419,966 training utterances (327 hours). We evaluate on a 9-speaker subset of the multi-speaker test data which contains 1808 utterances (comprising US, UK, Australian, and Indian speakers).

**Tasks**  The tasks that we explore include same-text prosody transfer, inter-text style transfer, and inter-speaker prosody transfer. We also evaluate the quality of samples produced when sampling via the prior. For these tasks, we compare performance when using variational models with and without the additional conditional dependencies in the variational posterior at a number of different capacity limits. For models with hierarchical latents, we demonstrate the effect of varying $C_H$ and $C_L$ for same-text prosody transfer when inferring $\mathbf{z}_H$ and sampling $\mathbf{z}_L$ or when inferring $\mathbf{z}_L$ directly.

**Evaluation**  We use crowd-sourced native speakers to collect two types of subjective evaluations. First, mean opinion score (MOS) rates naturalness on a scale of 1-5, 5 being the best. Second, we use the AXY side-by-side comparison proposed by Skerry-Ryan et al. (2018) to measure subjective similarity to a reference utterance relative to the baseline model on a scale of [-3,3]. For example, a score of 3 would mean that, compared to the baseline model, the model being tested produces samples much more perceptually similar to the ground truth reference. We also use an objective similarity metric that uses dynamic time warping to find the minimum mel cepstral distortion (Kubichek, 1993) between two sequences (MCD-DTW). Lastly, for inter-speaker transfer, we follow Skerry-Ryan et al. (2018) and use a simple speaker classifier to measure how well speaker identity is preserved. Additional details on evaluation methodologies are provided in Appendix A.

### 4.3 RESULTS

**Single speaker**  For single-speaker models, we compare the performance on same and inter-text transfer and the quality of samples generated via the prior for models with and without text conditioning in the variational posterior (*Var+Txt* and *Var*, respectively) at different capacity limits, $C$.

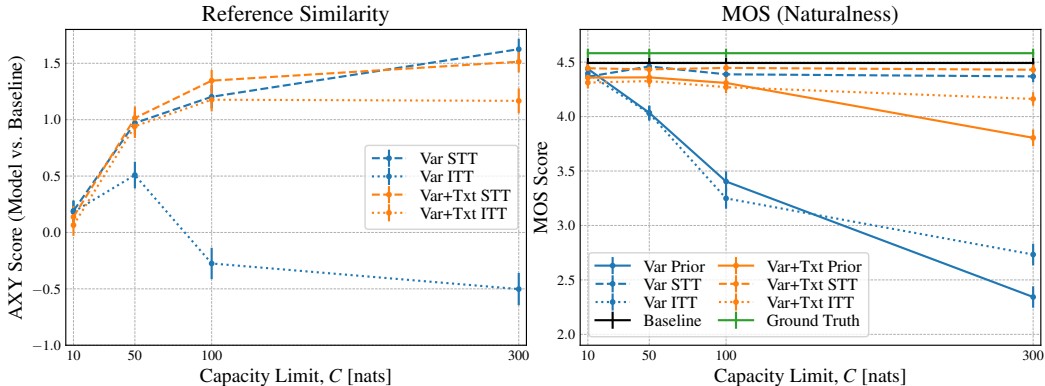

Figure 4: Comparing same-text transfer (STT), inter-text transfer (ITT), and prior samples (Prior) for variational models with and without text dependencies in the variational posterior (Var+Txt and Var, respectively). Error bars show 95% confidence intervals for the subjective evaluations.

Similarity results for the transfer task are shown on the left side of Figure 4 and demonstrate increasing reference similarity as $C$ is increased, with the exception of the model without text conditioning on the inter-text transfer task. Looking at the MOS naturalness results on the right side of Figure 4, we see that both inter-text transfer and prior sampling take a serious hit as capacity is increased for the Var model, while the Var+Txt model is able to maintain respectable performance even at very high capacities on all tasks.

Listening to the audio examples, we can hear that the Var model produces degenerate output at high capacities when attempting to transfer the style from a short utterance to a long utterance. This indicates that the decoder probably hasn't seen similar embeddings paired with long utterances during training, which suggests that $\mathbf{z}$ is improperly correlated with text length. Similar behavior is also observed when generating prior samples using shorter or longer text. This means that an arbitrary $\mathbf{z}$ (sampled from the prior or inferred from a reference) is unlikely to pair well with text of an arbitrary length.

**Multi-speaker** For multi-speaker models, we compare inter-speaker same-text transfer performance and prior sample quality with and without speaker conditioning in the variational posterior (*Var+Txt+Spk* and *Var+Txt*, respectively) at a fixed capacity limit of 150 nats. In Table 1, we see that both models are able to preserve characteristics of the reference utterance during transfer (AXY Ref. Similarity column), while the Var+Txt+Spk model has an edge in MOS for both inter-speaker transfer and prior samples (almost matching the MOS of the deterministic baseline model even at high embedding capacity).

Similar to the utterance length argument in the single speaker section above, it is likely that adding speaker dependencies to the posterior allows the model to use the entire latent space for each speaker (meaning $\mathbf{z}$ is not correlated with speaker identity), thereby forcing the decoder to learn to map all plausible points in the latent space to natural-sounding utterances that preserve the target speaker's pitch range. The speaker classifier results show that the Var+Txt+Spk model preserves target speaker identity about as well as the baseline model and ground truth data (~5% of the time the classifier chooses a speaker other than the target speaker), whereas for the Var+Txt model this happens about 22% of the time. Though 22% seems like a large speaker error rate, it is much lower than the 79% figure presented by Skerry-Ryan et al. (2018) for a heuristic prosody transfer model. This demonstrates that even with a weakly conditioned posterior, the capacity limiting properties of variational models lead to better transfer generality and robustness.

**Hierarchical latents** To evaluate hierarchical decomposition of capacity in a single speaker setting, we use the MCD-DTW distance to quantify reference similarity and same-reference inter-sample variability. As shown in Table B.1 in the appendix, MCD-DTW strongly (negatively) correlates with subjective similarity.

The left side of Figure 5 shows results for samples generated using high-level latents, $\mathbf{z}_{\mathrm{H}}$, inferred from the reference. As $C_{\mathrm{H}}$ is increased, we see a strong downward trend in the average distance to

Table 1: Inter-speaker same-text prosody transfer results for $C = 150$ with and without speaker dependencies in the variational posterior (Var+Txt+Spk and Var+Txt, respectively). SpkID denotes the fraction of the time the target speaker was chosen by the speaker classifier. For reference, we provide MOS and SpkID numbers for the baseline model and ground truth audio (though neither are "prior" samples).

| Model | Inter-Speaker Transfer | | | Prior Samples | |
| --- | --- | --- | --- | --- | --- |
| | AXY Ref. Similarity | MOS | SpkID | MOS | SpID |
| Var+Txt | $0.364 \pm 0.104$ | $3.994 \pm 0.066$ | 80.1% | $3.674 \pm 0.077$ | 78.0% |
| Var+Txt+Spk | $0.439 \pm 0.087$ | $4.099 \pm 0.061$ | 95.8% | $3.906 \pm 0.066$ | 94.9% |
| Baseline | - | - | - | $4.086 \pm 0.060$ | 95.7% |
| Ground Truth | - | - | - | $4.535 \pm 0.044$ | 96.9% |

the reference. We can also see that for a fixed $C_\mathrm{H}$, increasing $C_\mathrm{L}$ results in a larger amount of sample-to-sample variation (average MCD-DTW between samples) when inferring a single $\mathbf{z}_\mathrm{H}^\mathrm{ref}$ from the variational posterior and then sampling $\mathbf{z}_\mathrm{L} \sim p(\mathbf{z}_\mathrm{L}|\mathbf{z}_\mathrm{H}^\mathrm{ref})$ from the prior to use in the reconstructions.

The right side of Figure 5 shows the same metrics but for samples generated using low-level latents, $\mathbf{z}_\mathrm{L}^\mathrm{ref}$, inferred from the variational posterior. In this case, we see a slight downward trend in the reference distance as the total capacity limit, $C$, is increased (the trend is less dramatic because the capacity is already fairly high). We also see significantly lower inter-sample distance because the variation modeled by the latents is completely specified by $\mathbf{z}_\mathrm{L}$. In this case, we sample multiple $\mathbf{z}_\mathrm{L}^\mathrm{ref}$'s from $q(\mathbf{z}_\mathrm{L}|\mathbf{x}^\mathrm{ref}, \mathbf{y}_\mathrm{T}^\mathrm{ref})$ for the same $\mathbf{x}^\mathrm{ref}$ because using the same $\mathbf{z}_\mathrm{L}$ would lead to identical output from the deterministic decoder.

Using Capacitron with hierarchical latents increases the model's versatility for transfer tasks. By inferring just the high-level latents, $\mathbf{z}_\mathrm{H}$, from a reference, we can sample multiple realizations of an utterance that are similar to the reference, with the level of similarity controlled by $C_\mathrm{H}$, and the amount of sample-to-sample variation controlled by $C_\mathrm{L}$. The same model can also be used for higher fidelity, lower variability transfer by inferring the low-level latents, $\mathbf{z}_\mathrm{L}$, from a reference, with the level of similarity controlled by $C = C_\mathrm{H} + C_\mathrm{L}$. As mentioned before, this idea could also be extended to use additional levels of latents, thereby increasing transfer and sampling flexibility.

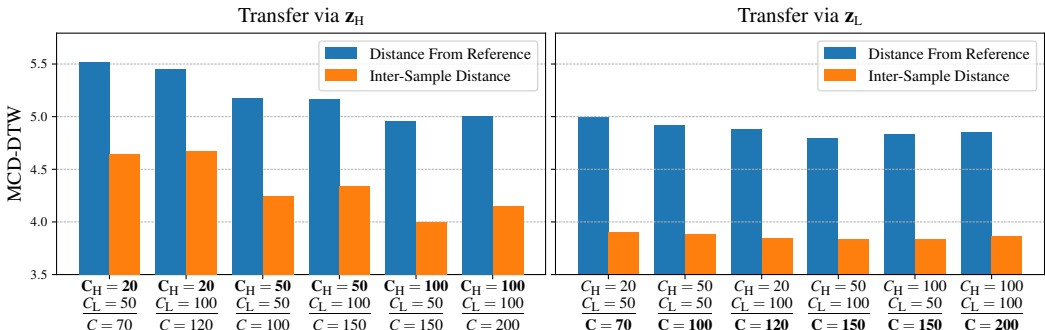

Figure 5: MCD-DTW reference distance and inter-sample distance for hierarchical latents when transferring via $\mathbf{z}_\mathrm{H}$ and $\mathbf{z}_\mathrm{L}$.

To appreciate the results fully, it is strongly recommended to listen to the audio examples available on the web[3].

---

[3] https://variational-embedding-capacity.github.io/demos/

## 5 CONCLUSION

We have proposed embedding capacity (i.e., representational mutual information) as a useful framework for comparing and configuring latent variable models of speech. Our proposed model, Capacitron, demonstrates that including text and speaker dependencies in the variational posterior allows a single model to be used successfully for a variety of transfer and sampling tasks. Motivated by the multi-faceted variability of natural human speech, we also showed that embedding capacity can be decomposed hierarchically in order to enable the model to control a trade-off between transfer fidelity and sample-to-sample variation.

There are many directions for future work, including adapting the fixed-length variational embeddings to be variable-length and synchronous with either the text or audio, using more powerful distributions like normalizing flows, and replacing the deterministic decoder with a proper likelihood distribution. For transfer and control uses cases, the ability to distribute certain speech characteristics across specific subsets of the hierarchical latents would allow more fine-grained control of different aspects of the output speech. And for purely generative, non-transfer use cases, using more powerful conditional priors could improve sample quality.

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

## A  Experiment details

### A.1  Architecture details

**Baseline Tacotron**  The baseline Tacotron we start with (which serves as $f_\theta(\cdot)$ in eq. (1)) is similar to the original sequence-to-sequence model described by Wang et al. (2017) but uses some modifications introduced by Skerry-Ryan et al. (2018). Input to the model consists of sequences of phonemes produced by a text normalization pipeline rather than character inputs. The CBHG text encoder from Wang et al. (2017) is used to convert the input phonemes into a sequence of text embeddings. Before being fed to the CBHG encoder, the phoneme inputs are converted to learned 256-dimensional embeddings and passed through a pre-net composed of two fully connected ReLU layers (with 256 and 128 units, respectively), with dropout of 0.5 applied to the output of each layer. For multi-speaker models, a learned embedding for the target speaker is broadcast-concatenated to the output of the text encoder.

The attention module uses a single LSTM layer with 256 units and zoneout of 0.1 followed by an MLP with 128 tanh hidden units to compute parameters for the monotonic 5-component GMM attention window. Instead of using the exponential function to compute the shift and scale parameters of the GMM components as in (Graves, 2013), we use the softplus function, which we found leads to faster alignment and more stable optimization.

The autoregressive decoder module consists of 2 LSTM layers each with 256 units, zoneout of 0.1, and residual connections between the layers. The spectrogram output is produced using a linear layer on top of the 2 LSTM layers, and we use a reduction factor of 2, meaning we predict two spectrogram frames for each decoder step. The decoder is fed the last frame of its most recent prediction (or the previous ground truth frame during training) and the current context as computed by the attention module. Before being fed to the decoder, the previous prediction is passed through the same pre-net used before the text encoder above.

**Mel spectrograms**  The mel spectrograms the model predicts are computed from 24kHz audio using a frame size of 50ms, a hop size of 12.5ms, an FFT size of 2048, and a Hann window. From the FFT energies, we compute 80 mel bins distributed between 80Hz and 12kHz.

**Reference encoder**  The common reference encoder we use to compute reference embeddings starts with the mel spectrogram from the reference and passes it through a stack of 6 convolutional layers, each using ReLU non-linearities, 3x3 filters, 2x2 stride, and batch normalization. The 6 layers have 32, 32, 64, 64, 128, and 128 filters, respectively. The output of this convolution stack is fed into a unidirectional LSTM with 128 units, and the final output of the LSTM serves as the output of our basic reference encoder.

To replicate the prosody transfer model from Skerry-Ryan et al. (2018), we pass the reference encoder output through an additional tanh bottleneck layer to compute the embedding. For the Style Tokens model in Wang et al. (2018), we pass the output through the Style Tokens bottleneck described in the paper. For the approximate posterior in our variational models, we pass the output of the reference encoder (and potentially vectors describing the text and/or speaker) through an MLP with 128 tanh hidden units to produce the parameters of the diagonal Gaussian posterior which we sample from to produce a reference embedding. For all models with reference encoders, the resulting reference embedding is broadcast-concatenated to the output of the text encoder.

**Conditional inputs**  When providing information about the text to the variational posterior, we pass the sequence of text embeddings produced by the text encoder to a unidirectional RNN with 128 units and use its final output as a fixed-length text summary that is passed to the posterior MLP. Speaker information is passed to the posterior MLP via a learned speaker embedding.

### A.2  Training Details

For the optimization problems shown in eqs. (9) and (14), we use two separate optimizers. The first minimizes the objective with respect to the model parameters using the SyncReplicasOptimizer [4] from Tensorflow with 10 workers (2 of them backup workers) and an effective batch size of 256. We

---

[4]`https://www.tensorflow.org/api_docs/python/tf/train/`
`SyncReplicasOptimizer`

also use gradient clipping with a threshold of 5. This optimizer uses the Adam algorithm (Kingma & Ba, 2015) with $\beta_1 = 0.9$, $\beta_2 = 0.999$, $\epsilon = 10^{-8}$, and a learning rate that is set to $10^{-3}$, $5 \times 10^{-4}$, $3 \times 10^{-4}$, $10^{-4}$, and $5 \times 10^{-5}$ at 50k, 100k, 150k, and 200k steps, respectively. Training is run for 300k steps total.

The optimizer that maximizes the objective with respect to the Lagrange multiplier is run asynchronously across the 10 workers (meaning each worker computes an independent update using its 32-example sub-batch) and uses SGD with a momentum of 0.9 and a learning rate of $10^{-5}$. The Lagrange multiplier is computed by passing an unconstrained parameter through the softplus function in order to enforce non-negativity. The initial value of the parameter is chosen such that the Lagrange multiplier equals 1 at the start of training.

### A.3 EVALUATION DETAILS

**Subjective evaluation**  Details for the subjective reference similarity and MOS naturalness evaluations are provided in Figures A.1 and A.2. To evaluate reference similarity, we use the AXY side-by-side template in Figure A.1, where A is the reference utterance, and X and Y are outputs from the model being tested and the baseline model.

**MCD-DTW**  We evaluate the models with hierarchical latents using the MCD-DTW distance to quantify reference similarity and the amount of inter-sample variation. To compute mel cepstral distortion (MCD) (Kubichek, 1993), we use the same mel spectrogram parameters described in A.1 and take the DCT to compute the first 13 MFCCs (not including the 0th coefficient). The MCD between two frames is the Euclidean distance between their MFCC vectors. Then we use the dynamic time warping (DTW) algorithm (Müller, 2007) (with a warp penalty of 1.0) to find an alignment between two spectrograms that produces the minimum MCD cost (including the total warp penalty). We report the average per-frame MCD-DTW.

To evaluate reference similarity, we simply compute the MCD-DTW between the synthesized audio and the reference audio (a lower MCD-DTW indicates higher similarity). The strong (negative) correlation between MCD-DTW and subjective similarity is demonstrated in Table B.1. To quantify inter-sample variation, we compute 5 output samples using the same reference and compute the average MCD-DTW between the first sample and each subsequent sample.

## B    ADDITIONAL RESULTS

**Rate-distortion plots**  In Figure B.1, we augment the reconstruction loss plots from Figure 1 with additional rate/distortion plots (Alemi et al., 2018) and vary the KL weight, $\beta$, and as well as $C$.

**Optimization Examples**  Figure B.2 shows examples of how the KL weight, $\beta$, and KL term, $R$, evolve during training for different capacity limits, $C$. The curves shown are from the single speaker Var+Txt models discussed in Section 4.3. The target KL is achieved very quickly and then maintained throughout training via continual updates to $\beta$ that are interleaved with updates to the model parameters.

**Single-speaker similarity and naturalness results**  Tables B.1 and B.2 list the raw numbers used in the single-speaker reference similarity and MOS naturalness plots shown in Figure 4 in the main paper. Also shown is MCD-DTW reference distance alongside subjective reference similarity.

**Hierarchical latents results**  The similarity and inter-sample variability results for hierarchical latents from Figure 5 are shown in table format in Table B.3.

Figure A.1: Evaluation template for AXY prosodic reference similarity side-by-side evaluations. A human rater is presented with three stimuli: a reference speech sample (A), and two competing samples (X and Y) to evaluate. The rater is asked to rate whether the prosody of X or Y is closer to that of the reference on a 7-point scale. The scale ranges from "X is much closer" to "Both are about the same distance" to "Y is much closer", and can naturally be mapped on the integers from -3 to 3. Prior to collecting any ratings, we provide the raters with 4 examples of prosodic attributes to evaluate (intonation, stress, speaking rate, and pauses), and explicitly instruct the raters to ignore audio quality or pronunciation differences. For each triplet (A, X, Y) evaluated, we collect 1 rating, and no rater is used for more than 6 items in a single evaluation. To analyze the data from these subjective tests, we average the scores and compute 95% confidence intervals.

Table B.1: Detailed subjective reference similarity scores and objective MCD-DTW reference distance for single speaker models at different capacity limits, $C$, and with and without text conditioning in the variational posterior (Var+Txt and Var, respectively). Notice how subjective reference similarity for same-text transfer is strongly negatively correlated with MCD-DTW.

| | Ref. Similarity | | MCD-DTW |
|---|---|---|---|
| Model | Same-TT | Inter-TT | Same-TT |
| Var($C$=10) | $0.192 \pm 0.093$ | $0.182 \pm 0.097$ | 5.67 |
| Var($C$=50) | $0.970 \pm 0.102$ | $0.509 \pm 0.118$ | 5.13 |
| Var($C$=100) | $1.203 \pm 0.102$ | $-0.275 \pm 0.139$ | 5.04 |
| Var($C$=300) | $1.625 \pm 0.092$ | $-0.502 \pm 0.143$ | 4.81 |
| Var+Txt($C$=10) | $0.138 \pm 0.097$ | $0.065 \pm 0.095$ | 5.68 |
| Var+Txt($C$=50) | $1.014 \pm 0.102$ | $0.942 \pm 0.104$ | 5.11 |
| Var+Txt($C$=100) | $1.346 \pm 0.096$ | $1.177 \pm 0.103$ | 4.94 |
| Var+Txt($C$=300) | $1.514 \pm 0.095$ | $1.167 \pm 0.110$ | 4.83 |

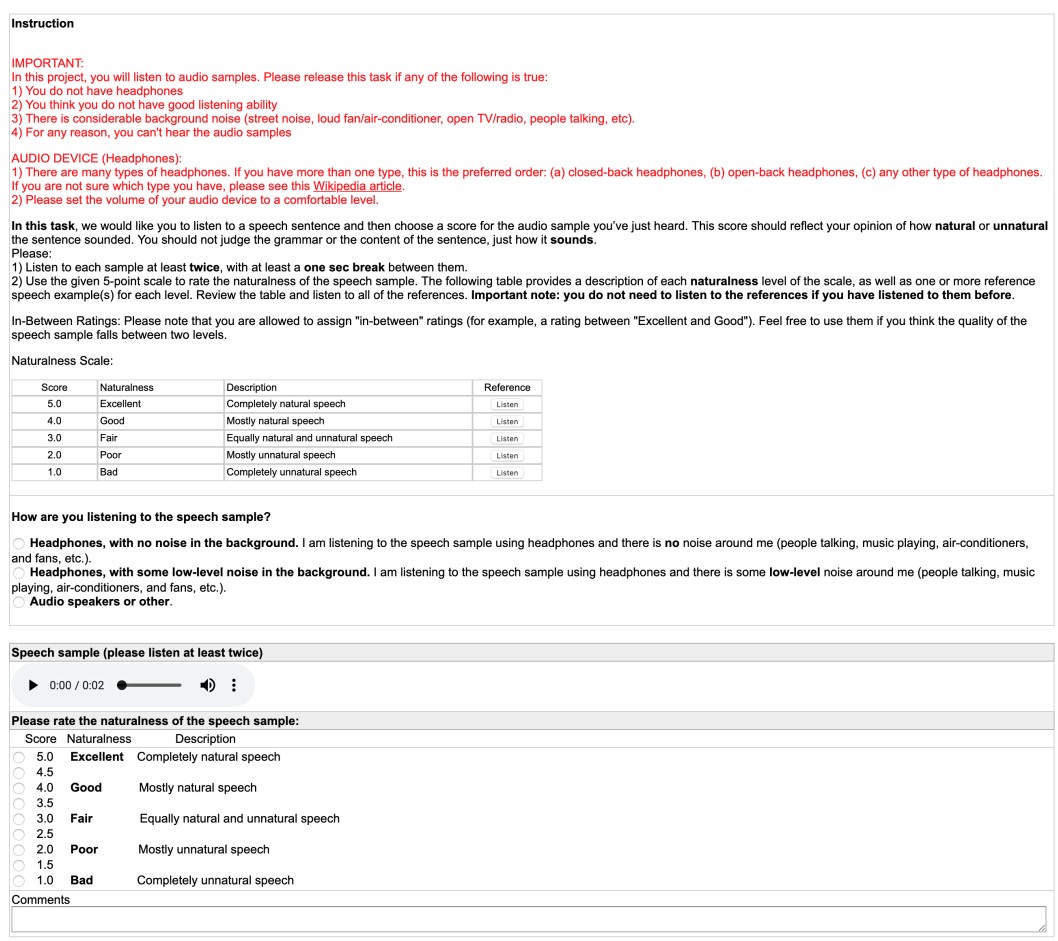

Figure A.2: Evaluation template for mean opinion score (MOS) naturalness ratings. A human rater is presented with a single speech sample and is asked to rate perceived naturalness on a scale of 1–5, where 1 is "Bad" and 5 is "Excellent". For each sample, we collect 1 rating, and no rater is used for more than 6 items in a single evaluation. To analyze the data from these subjective tests, we average the scores and compute 95% confidence intervals. Natural human speech is typically rated around 4.5.

Table B.2: MOS naturalness scores for single speaker models at different capacity limits, $C$, with and without text conditioning in the variational posterior (Var+Txt and Var, respectively). Scores are shown for prior samples (Prior), same-text transfer (Same-TT), and inter-text transfer (Inter-TT). These results are visualized in Figure 4 in the main paper.

| Model | MOS score | | |
| --- | --- | --- | --- |
| | Prior | Same-TT | Inter-TT |
| Ground Truth | $4.582 \pm 0.041$ | | |
| Base | $4.492 \pm 0.048$ | | |
| Var($C$=10) | $4.438 \pm 0.049$ | $4.366 \pm 0.053$ | $4.396 \pm 0.049$ |
| Var($C$=50) | $4.035 \pm 0.066$ | $4.460 \pm 0.051$ | $4.029 \pm 0.067$ |
| Var($C$=100) | $3.404 \pm 0.093$ | $4.388 \pm 0.055$ | $3.249 \pm 0.095$ |
| Var($C$=300) | $2.343 \pm 0.098$ | $4.369 \pm 0.054$ | $2.733 \pm 0.099$ |
| Var+Txt($C$=10) | $4.358 \pm 0.056$ | $4.444 \pm 0.052$ | $4.312 \pm 0.053$ |
| Var+Txt($C$=50) | $4.360 \pm 0.053$ | $4.433 \pm 0.052$ | $4.326 \pm 0.054$ |
| Var+Txt($C$=100) | $4.309 \pm 0.056$ | $4.447 \pm 0.048$ | $4.270 \pm 0.055$ |
| Var+Txt($C$=300) | $3.805 \pm 0.076$ | $4.430 \pm 0.050$ | $4.162 \pm 0.062$ |

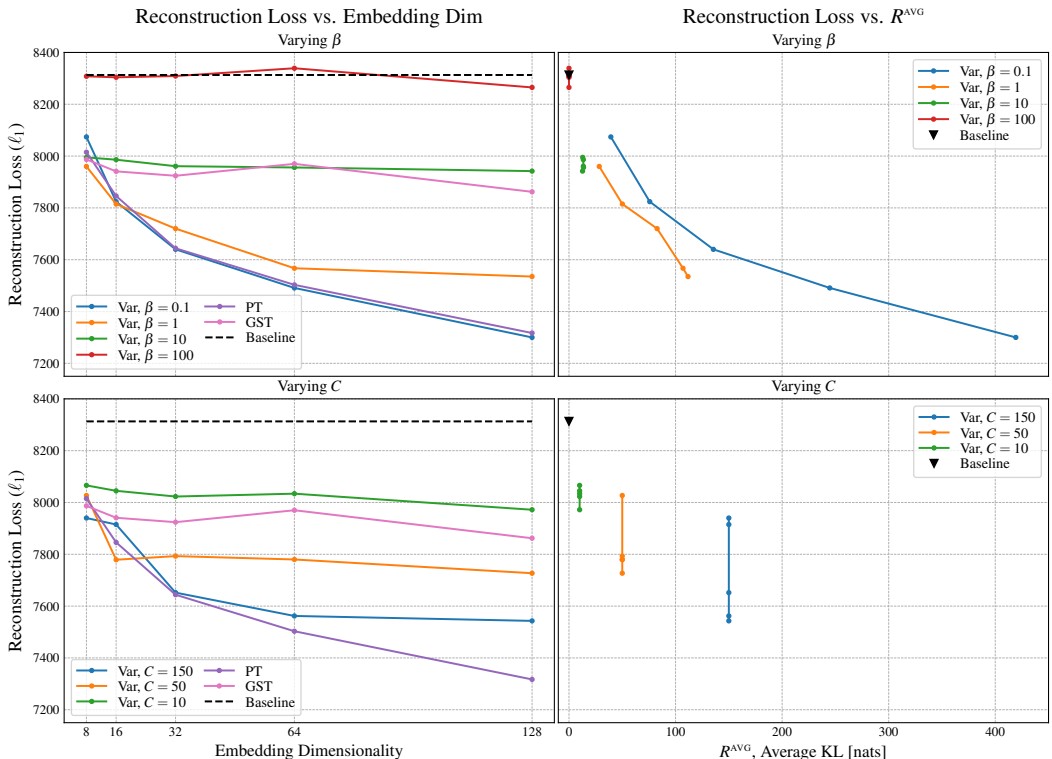

Figure B.1: Figure 1 with additional plots showing reconstruction loss vs. the average KL term. In these plots we can see how $R^{\text{AVG}}$ varies with embedding dimensionality for constant KL weight, $\beta$, while using the KL limit, $C$, from the optimization problem in eq. 9 achieves constant $R^{\text{AVG}}$.

Table B.3: Transfer using hierarchical latents. "Ref." is the the average MCD-DTW distance from the reference, and "X-samp." is the average inter-sample MCD-DTW. These are the numbers used in the plots in Figure 5.

(a) Transfer via $\mathbf{z}_{\text{H}}$.

| Capacity Limits | | | MCD-DTW | |
|---|---|---|---|---|
| $\mathbf{C}_{\text{H}}$ | $C_{\text{L}}$ | $C$ | Ref. | X-samp. |
| **0** | 0 | 0 | 6.054 | - |
| **20** | 50 | 70 | 5.517 | 4.638 |
| **20** | 100 | 120 | 5.453 | 4.670 |
| **50** | 50 | 100 | 5.172 | 4.245 |
| **50** | 100 | 150 | 5.166 | 4.332 |
| **100** | 50 | 150 | 4.952 | 3.999 |
| **100** | 100 | 200 | 5.000 | 4.147 |

(b) Transfer via $\mathbf{z}_{\text{L}}$.

| Capacity Limits | | | MCD-DTW | |
|---|---|---|---|---|
| $C_{\text{H}}$ | $C_{\text{L}}$ | $\mathbf{C}$ | Ref. | X-samp. |
| 0 | 0 | **0** | 6.054 | - |
| 20 | 50 | **70** | 4.991 | 3.899 |
| 50 | 50 | **100** | 4.916 | 3.876 |
| 20 | 100 | **120** | 4.882 | 3.847 |
| 100 | 50 | **150** | 4.834 | 3.830 |
| 50 | 100 | **150** | 4.797 | 3.832 |
| 100 | 100 | **200** | 4.852 | 3.858 |

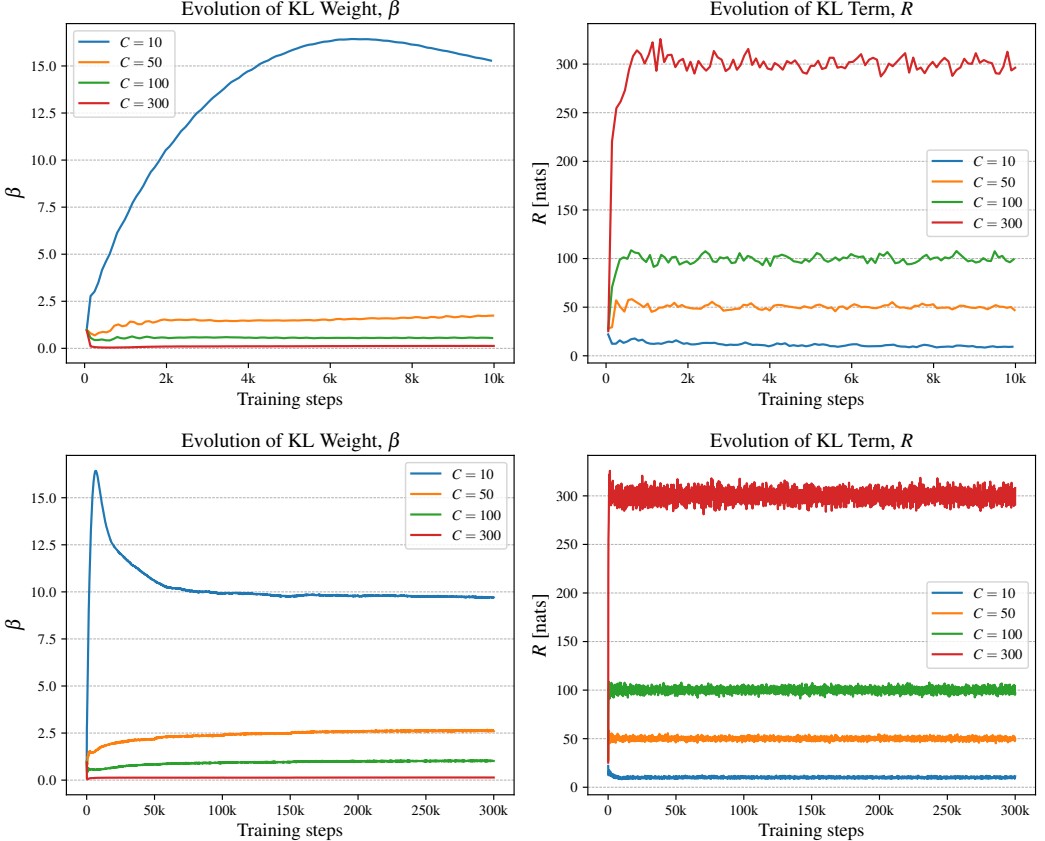

Figure B.2: Evolution of the KL weight, $\beta$, and KL term, $R$, for different capacity limits, $C$. The values for the KL term are computed using single training batches. The top plots show the first 10,000 training steps, while the bottom plots show the entire 300,000-step training run. The target KL is achieved within the first 1,000 steps and then maintained throughout training via continual updates to $\beta$ that are interleaved with updates to the model parameters. The value of $\beta$ is initialized at 1.0 and ends up at around 9.7, 2.6, 1.0, and 0.14 for $C$ = 10, 50, 100, and 300, respectively.

# C DERIVATIONS

## C.1 BOUNDING REPRESENTATIONAL MUTUAL INFORMATION

Definitions:

$$R \equiv \int q(\mathbf{z}|\mathbf{x}) \log \frac{q(\mathbf{z}|\mathbf{x})}{p(\mathbf{z})} d\mathbf{z} \qquad \text{(KL term)} \qquad (15)$$

$$R^{\text{AVG}} \equiv \iint p_D(\mathbf{x}) q(\mathbf{z}|\mathbf{x}) \log \frac{q(\mathbf{z}|\mathbf{x})}{p(\mathbf{z})} d\mathbf{x} d\mathbf{z} \qquad \text{(Average KL term)} \qquad (16)$$

$$I_q(\mathbf{X}; \mathbf{Z}) \equiv \iint p_D(\mathbf{x}) q(\mathbf{z}|\mathbf{x}) \log \frac{q(\mathbf{z}|\mathbf{x})}{q(\mathbf{z})} d\mathbf{x} d\mathbf{z} \qquad \text{(Representational mutual information)} \qquad (17)$$

$$q(\mathbf{z}) \equiv \int p_D(\mathbf{x}) q(\mathbf{z}|\mathbf{x}) d\mathbf{x} \qquad \text{(Aggregated posterior)} \qquad (18)$$

KL non-negativity:

$$\int q(x) \log \frac{q(x)}{p(x)} dx \geq 0 \qquad (19)$$

$$\implies \int q(x) \log q(x) \geq \int q(x) \log p(x) dx \qquad (20)$$

Mutual information is upper bounded by the average KL (Alemi et al., 2018):

$$I_q(\mathbf{X}; \mathbf{Z}) \equiv \iint p_D(\mathbf{x}) q(\mathbf{z}|\mathbf{x}) \log \frac{q(\mathbf{z}|\mathbf{x})}{q(\mathbf{z})} d\mathbf{x} d\mathbf{z} \qquad (21)$$

$$= \iint p_D(\mathbf{x}) q(\mathbf{z}|\mathbf{x}) \log q(\mathbf{z}|\mathbf{x}) d\mathbf{x} d\mathbf{z} - \iint p_D(\mathbf{x}) q(\mathbf{z}|\mathbf{x}) \log q(\mathbf{z}) d\mathbf{x} d\mathbf{z} \qquad (22)$$

$$= \iint p_D(\mathbf{x}) q(\mathbf{z}|\mathbf{x}) \log q(\mathbf{z}|\mathbf{x}) d\mathbf{x} d\mathbf{z} - \int q(\mathbf{z}) \log q(\mathbf{z}) d\mathbf{z} \qquad (23)$$

$$\leq \iint p_D(\mathbf{x}) q(\mathbf{z}|\mathbf{x}) \log q(\mathbf{z}|\mathbf{x}) d\mathbf{x} d\mathbf{z} - \int q(\mathbf{z}) \log p(\mathbf{z}) d\mathbf{z} \qquad (24)$$

$$= \iint p_D(\mathbf{x}) q(\mathbf{z}|\mathbf{x}) \log q(\mathbf{z}|\mathbf{x}) d\mathbf{x} d\mathbf{z} - \iint p_D(\mathbf{x}) q(\mathbf{z}|\mathbf{x}) \log p(\mathbf{z}) d\mathbf{x} d\mathbf{z} \qquad (25)$$

$$= \iint p_D(\mathbf{x}) q(\mathbf{z}|\mathbf{x}) \log \frac{q(\mathbf{z}|\mathbf{x})}{p(\mathbf{z})} d\mathbf{x} d\mathbf{z} \qquad (26)$$

$$\equiv R^{\text{AVG}} \qquad (27)$$

$$\implies I_q(\mathbf{X}; \mathbf{Z}) \leq R^{\text{AVG}} \qquad (28)$$

where the inequality in (24) follows from (20).

The difference between the average KL and the mutual information is the aggregate KL:

$$R^{\text{AVG}} - I_q(\mathbf{X}; \mathbf{Z}) = \iint p_D(\mathbf{x}) q(\mathbf{z}|\mathbf{x}) \log \frac{q(\mathbf{z})}{p(\mathbf{z})} d\mathbf{x} d\mathbf{z} \qquad (29)$$

$$= \int q(\mathbf{z}) \log \frac{q(\mathbf{z})}{p(\mathbf{z})} d\mathbf{z} \qquad (30)$$

$$= D_{\text{KL}}(q(\mathbf{z})\|p(\mathbf{z})) \qquad \text{(Aggregate KL)} \qquad (31)$$

## C.2 HIERARCHICALLY BOUNDING MUTUAL INFORMATION

The model with hierarchical latents shown in Figure C.1 gives us the following:

$$p(\mathbf{z}) = p(\mathbf{z}_{\text{H}}, \mathbf{z}_{\text{L}}) = p(\mathbf{z}_{\text{L}}|\mathbf{z}_{\text{H}}) p(\mathbf{z}_{\text{H}}) \qquad (32)$$

$$q(\mathbf{z}|\mathbf{x}) = q(\mathbf{z}_{\text{H}}, \mathbf{z}_{\text{L}}|\mathbf{x}) = q(\mathbf{z}_{\text{L}}|\mathbf{x}) q(\mathbf{z}_{\text{H}}|\mathbf{z}_{\text{L}}) \qquad (33)$$

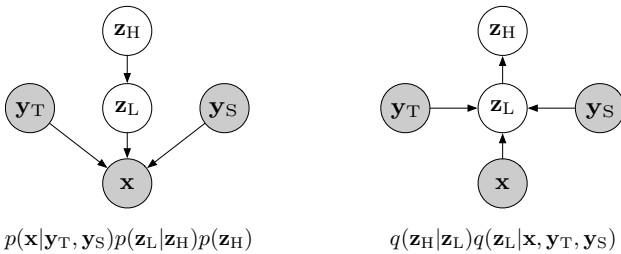

$$p(\mathbf{x}|\mathbf{y}_{\mathrm{T}}, \mathbf{y}_{\mathrm{S}})p(\mathbf{z}_{\mathrm{L}}|\mathbf{z}_{\mathrm{H}})p(\mathbf{z}_{\mathrm{H}}) \qquad\qquad q(\mathbf{z}_{\mathrm{H}}|\mathbf{z}_{\mathrm{L}})q(\mathbf{z}_{\mathrm{L}}|\mathbf{x}, \mathbf{y}_{\mathrm{T}}, \mathbf{y}_{\mathrm{S}})$$

Figure C.1: Hierarchical decomposition of the latents. Shaded nodes indicate observed variables. [left] The true generative model. [right] Variational posterior that matches the form of the true posterior.

The conditional dependencies on $\mathbf{y}_{\mathrm{T}}$ and $\mathbf{y}_{\mathrm{S}}$ are omitted for compactness.

Define marginal aggregated posteriors:

$$q(\mathbf{z}_{\mathrm{L}}) \equiv \int p_D(\mathbf{x})q(\mathbf{z}_{\mathrm{L}}|\mathbf{x})d\mathbf{x} \tag{34}$$

$$q(\mathbf{z}_{\mathrm{H}}) \equiv \int q(\mathbf{z}_{\mathrm{L}})q(\mathbf{z}_{\mathrm{H}}|\mathbf{z}_{\mathrm{L}})d\mathbf{z}_{\mathrm{L}} \tag{35}$$

We can write the average joint KL term and mutual information as follows:

$$R^{\mathrm{AVG}} = \int p_D(\mathbf{x})[D_{\mathrm{KL}}(q(\mathbf{z}_{\mathrm{H}}|\mathbf{z}_{\mathrm{L}})q(\mathbf{z}_{\mathrm{L}}|\mathbf{x})\|p(\mathbf{z}_{\mathrm{L}}|\mathbf{z}_{\mathrm{H}})p(\mathbf{z}_{\mathrm{H}}))]d\mathbf{x} \tag{36}$$

$$I_q(\mathbf{X}; [\mathbf{Z}_{\mathrm{H}}, \mathbf{Z}_{\mathrm{L}}]) = \int p_D(\mathbf{x})[D_{\mathrm{KL}}(q(\mathbf{z}_{\mathrm{H}}|\mathbf{z}_{\mathrm{L}})q(\mathbf{z}_{\mathrm{L}}|\mathbf{x})\|q(\mathbf{z}_{\mathrm{H}}|\mathbf{z}_{\mathrm{L}})q(\mathbf{z}_{\mathrm{L}}))]d\mathbf{x} \tag{37}$$

Next we show that $I_q(\mathbf{X}; [\mathbf{Z}_{\mathrm{H}}, \mathbf{Z}_{\mathrm{L}}]) = I_q(\mathbf{X}; \mathbf{Z}_{\mathrm{L}})$:

$$I_q(\mathbf{X}; [\mathbf{Z}_{\mathrm{H}}, \mathbf{Z}_{\mathrm{L}}]) = \iiint p_D(\mathbf{x})q(\mathbf{z}_{\mathrm{H}}, \mathbf{z}_{\mathrm{L}}|\mathbf{x}) \log \frac{q(\mathbf{z}_{\mathrm{H}}|\mathbf{z}_{\mathrm{L}})q(\mathbf{z}_{\mathrm{L}}|\mathbf{x})}{q(\mathbf{z}_{\mathrm{H}}|\mathbf{z}_{\mathrm{L}})q(\mathbf{z}_{\mathrm{L}})}d\mathbf{x}d\mathbf{z}_{\mathrm{H}}d\mathbf{z}_{\mathrm{L}} \tag{38}$$

$$= \iiint p_D(\mathbf{x})q(\mathbf{z}_{\mathrm{H}}, \mathbf{z}_{\mathrm{L}}|\mathbf{x}) \log \frac{q(\mathbf{z}_{\mathrm{L}}|\mathbf{x})}{q(\mathbf{z}_{\mathrm{L}})}d\mathbf{x}d\mathbf{z}_{\mathrm{H}}\mathbf{z}_{\mathrm{L}} \tag{39}$$

$$= \iint p_D(\mathbf{x})q(\mathbf{z}_{\mathrm{L}}|\mathbf{x}) \log \frac{q(\mathbf{z}_{\mathrm{L}}|\mathbf{x})}{q(\mathbf{z}_{\mathrm{L}})}d\mathbf{x}d\mathbf{z}_{\mathrm{L}} \tag{40}$$

$$= I_q(\mathbf{X}; \mathbf{Z}_{\mathrm{L}}) \tag{41}$$

Bound $I_q(\mathbf{X}; \mathbf{Z}_{\mathrm{L}})$:

$$I_q(\mathbf{X}; [\mathbf{Z}_{\mathrm{H}}, \mathbf{Z}_{\mathrm{L}}]) = I_q(\mathbf{X}; \mathbf{Z}_{\mathrm{L}}) \tag{42}$$

$$I_q(\mathbf{X}; [\mathbf{Z}_{\mathrm{H}}, \mathbf{Z}_{\mathrm{L}}]) \leq R^{\mathrm{AVG}} \tag{43}$$

$$\implies I_q(\mathbf{X}; \mathbf{Z}_{\mathrm{L}}) \leq R^{\mathrm{AVG}} \tag{44}$$

where (43) was shown in eq. (28).

Again, using the non-negativity of the KL, we can bound $I_q(\mathbf{Z}_H; \mathbf{Z}_L)$:

$$I_q(\mathbf{Z}_H; \mathbf{Z}_L) = \iint q(\mathbf{z}_H|\mathbf{z}_L)q(\mathbf{z}_L) \log \frac{q(\mathbf{z}_H|\mathbf{z}_L)}{q(\mathbf{z}_H)} d\mathbf{z}_H d\mathbf{z}_L \tag{45}$$

$$\leq \iint q(\mathbf{z}_H|\mathbf{z}_L)q(\mathbf{z}_L) \log \frac{q(\mathbf{z}_H|\mathbf{z}_L)}{p(\mathbf{z}_H)} d\mathbf{z}_H d\mathbf{z}_L \tag{46}$$

$$= \iiint p_D(\mathbf{x})q(\mathbf{z}_H|\mathbf{z}_L)q(\mathbf{z}_L|\mathbf{x}) \log \frac{q(\mathbf{z}_H|\mathbf{z}_L)}{p(\mathbf{z}_H)} d\mathbf{z}_H d\mathbf{z}_L d\mathbf{x} \tag{47}$$

$$= \iint p_D(\mathbf{x})q(\mathbf{z}_L|\mathbf{x})D_{\text{KL}}(q(\mathbf{z}_H|\mathbf{z}_L)\|p(\mathbf{z}_H))d\mathbf{z}_L d\mathbf{x} \tag{48}$$

$$\equiv R_H^{\text{AVG}} \tag{49}$$

$$I_q(\mathbf{X}; \mathbf{Z}_H) \leq I_q(\mathbf{Z}_L; \mathbf{Z}_H) \tag{50}$$

$$\implies I_q(\mathbf{X}; \mathbf{Z}_H) \leq R_H^{\text{AVG}} \tag{51}$$

where (50) can be demonstrated by applying the data processing inequality to a reversed version of the Markov chain, $\mathbf{X} \to \mathbf{Z}_L \to \mathbf{Z}_H$

Define $R_L$:

$$R_L \equiv R - R_H \tag{52}$$

$$= \iint q(\mathbf{z}_L|\mathbf{x})q(\mathbf{z}_H|\mathbf{z}_L) \log \frac{q(\mathbf{z}_L|\mathbf{x})}{p(\mathbf{z}_L|\mathbf{z}_H)} d\mathbf{z}_H d\mathbf{z}_L \tag{53}$$

Giving us the following bounds on $I_q(\mathbf{X}; \mathbf{Z}_L)$ and $I_q(\mathbf{X}; \mathbf{Z}_H)$:

$$\implies I_q(\mathbf{X}; \mathbf{Z}_H) \leq R_H^{\text{AVG}}, \qquad I_q(\mathbf{X}; \mathbf{Z}_L) \leq R_H^{\text{AVG}} + R_L^{\text{AVG}} \tag{54}$$

