# OpenReview forum: "Effective Use of Variational Embedding Capacity in Expressive End-to-End Speech Synthesis"
_ICLR.cc/2020/Conference — Reject_

### Official Review · AnonReviewer2 · 2019-10-20
**Official Blind Review #2**

**Rating:** 6

**Review:**

1. Summary: This paper proposes Capacitron, a conditional variational latent variable model for TTS which allow for controllable latent variable capacity. They optimize the Lagrangian dual of the ELBO and restrict the capacity of the rate-term through a learnable, non-negative multiplier. They demonstrate the effectiveness of their approach on a range of TTS tasks such as same-text prosody transfer and inter-text style transfer, and provide extensive analyses on their latent variable capacity (in addition to comparisons to non-variational approaches based on Tacotron).

2. Decision: Weak accept. I recommend this paper for acceptance due to its strong empirical results and clear presentation of the approach/unification of existing methods.

3. Supporting arguments: The extension of Capacitron to existing methods such as [Hsu et al., 2019 and Zhang et al., 2019] is simple (basically adopting the beta-VAE approach), as the conditional generative model in both the vanilla and hierarchical forms exist already. But the authors do a thorough job evaluating the strengths and weaknesses of their method through ablation studies on latent variable capacity and comparing to existing baselines in their experiments. The results are also convincing, as demonstrated through human listening tests and the samples provided in the supplement.

4. Feedback:
- The authors mention in Section 3.1 that in previous work, the variational posterior has the form q(z|x). While this is true for [Zhang et. al, 2019], I believe that [Hsu et. al, 2019] also uses the conditional generative model as described in Figure 3 -- it would be helpful to provide a more clear distinction between the two works in the exposition. From what I understood, this work’s contribution is not so much the introduction of the conditional generative model but identifying the effects of controlling the rate term in the ELBO decomposition.
- In Section 3.2, there is a lot of notation and several terms that make it difficult to parse Eq. 14 at first glance. For example, (1) R_L is never explicitly written out, (2) and is R == R_avg? It would be helpful to clean up this section so that eyeballing Eq. 14 would be easier for the reader (e.g. having all the relevant terms organized).

5. Questions:
- In the “Single speaker” section of Section 4.3, you mentioned that the “model has to divide the latent space into regions that correspond to different utterance lengths.” I’m curious -- is this something that was observed empirically?

**Experience Assessment:**

I do not know much about this area.

**Review Assessment: Checking Correctness Of Derivations And Theory:**

I assessed the sensibility of the derivations and theory.

**Review Assessment: Checking Correctness Of Experiments:**

I assessed the sensibility of the experiments.

**Review Assessment: Thoroughness In Paper Reading:**

I read the paper at least twice and used my best judgement in assessing the paper.

---

> ### Author Response · Authors · 2019-11-12
> **Response to Review #2**
>
> Thank you for your thoughtful and thorough review.
>
> We'd like to mention that although [Hsu et al., 2019] can be interpreted as having hierarchical latents, that approach is implemented using a Gaussian mixture prior (where the mixture component index can be interpreted as a discrete high-level latent variable). Whereas in our work, we use a continuous vector for the high-level latents. Our approach can be extended to use additional levels of latents each with its own capacity constraint if desired. Both approaches have their merits, but they are structurally different models.
>
> It's true that both [Zhang et al., 2019] and [Hsu et al., 2019] are conditional generative models (because TTS models are inherently conditional models of audio given text); however, neither uses a variational posterior that contains the conditional dependencies present in the true posterior (i.e., they both use q(z|x)).  See the second sentence of Section 2.2 in [Hsu et al., 2019] at https://arxiv.org/pdf/1810.07217v2.pdf. We also contacted the authors to confirm this. So, while matching the form of the true posterior isn't a huge breakthrough, it is one of the contributions of this paper which helps us to achieve the presented results.  We'll try to clarify this in the revision.
>
> Thanks for pointing out the potential confusion leading up to eq. (14). We were hoping that readers could generalize the relation between R (the KL term for a single data point) and R^{AVG} (the KL term marginalized over the entire dataset) in eq. (5) to R_L/R_H.  We've modified the descriptions of eqs. (10-14) to hopefully make this more clear.
>
> To answer your question about the model dividing up the latent space, this is an untested hypothesis based on the observed behavior of the models when using shorter and longer utterances. For example, the Var+Txt model yields intelligible output when attempting to transfer the style from a short utterance to a long utterance, while the Var model typically yields degenerate output. This indicates that for the Var model, the decoder probably hasn't seen similar embeddings paired with longer text, which suggests z is improperly correlated with text length.  Similar behavior is also observed when generating prior samples using shorter or longer text.

---

> > ### Comment · AnonReviewer2 · 2019-11-14
> > **thanks!**
> >
> > I see -- thanks for your clarification regarding [Zhang et al., 2019] and [Hsu et al, 2019]. If there is room in the revised version to make the distinctions more clear (I didn't see it in the updated PDF), I think fleshing out these subtle distinctions may be a good idea for clarity. The new equation and the latent space portions were clear and I appreciate the authors taking them into account!

---

> > > ### Author Response · Authors · 2019-11-15
> > > **Added additional clarification about hierarchical latents**
> > >
> > > Thanks for the suggestion. We've added comments that should make it easier to compare and contrast the hierarchical approach in [Hsu et al, 2019] and ours.  Specifically, to the 5th paragraph of Section 1, and the 2nd paragraph of Section 3.2.

---

### Official Review · AnonReviewer1 · 2019-10-23
**Official Blind Review #1**

**Rating:** 6

**Review:**

This paper introduces a new embedding method in expressive TTS by focusing on the capacity of hidden variables introduced in variational autoencoder. Such a method is supported by the KL divergence and lower bound theory and the paper well formulates/describes the capacity concept. In the experiments, the proposed method is compared with other conventional methods with well-designed subjective and objective metrics and shows the effectiveness in terms of the style transfer. The paper is well written in general.

One of the difficulties in this paper is reproducibility. The paper does not seem to provide source code and part of the data used in this paper also does not seem to be public data like libriTTS (I would be wrong but I could read that the multispeaker training data are not public), although I appreciate the authors' efforts to provide the implementation and evaluation details as much as possible in the appendix.

Another discussion of this paper is that it is not explicit to provide the meaning of the latent variables (this is a general issue in VAE) and I could not be fully convinced by the discussion and analysis based on the latent variable. There would be several semi-supervised studies to explicitly connect some (elements) of latent variables with actual attributes and I'd like to ask the authors to consider such direction to make the latent variable discussion in the paper more plausible.


**Experience Assessment:**

I have published one or two papers in this area.

**Review Assessment: Checking Correctness Of Derivations And Theory:**

I did not assess the derivations or theory.

**Review Assessment: Checking Correctness Of Experiments:**

I assessed the sensibility of the experiments.

**Review Assessment: Thoroughness In Paper Reading:**

I read the paper at least twice and used my best judgement in assessing the paper.

---

> ### Author Response · Authors · 2019-11-12
> **Response to Review #1**
>
> Thank you for your comments, and we're glad you appreciated the formulation, experiments, and writing.
>
> We agree that reproducibility (in terms of code and datasets) is currently an issue with the field of TTS in general, which is why (as you noticed) we took the extra effort to explain the architectural and training details in the appendix.  However, our organization's policies currently prevent us from open-sourcing our code due to legal and ethical concerns. We chose to use private datasets in our evaluations because the quality and expressiveness of existing public datasets is lacking, making it harder to demonstrate prosody and style transfer; though it might've been wiser to run our experiments on a mix of public and private datasets for completeness. These are ongoing issues for the TTS community as a whole, which we hope will be resolved in the coming months due to an increased emphasis on reproducibility from the community as a whole.
>
> We did not touch on the meaning of individual dimensions of the latent embeddings in this paper because the focus was on reference-based prosody and style transfer (which is agnostic to how specific generative factors are distributed in the latent space).  We agree that a semi-supervised approach would be an effective way to imbue a subset of the latent dimensions with interpretable meanings, and we are actively pursuing this direction in followup work.

---

### Official Review · AnonReviewer3 · 2019-10-27
**Official Blind Review #3**

**Rating:** 3

**Review:**

In this work authors present a regularized, variational autoencoder method for speech synthesis. To endow the latent space with more capacity, the authors employ a modified variational autoencoder objective, which uses a learnable Lagrange multiplier to impose a capacity limit on KL divergence between latent posterior and prior. The authors furthermore propose to decompose the latent embedding space into a two-level hierarchical representation to give generative process more control over style transfer and sample-to-sample variance. They extend earlier theoretical results providing upper bounds on the mutual information between data and its latent embedding to their hierarchical latent representation. In numerical experiments the authors evaluate their approach on a number of speech synthesis tasks involving same-text prosody transfer, inter-text style transfer, inter-speaker prosody transfer. They also analyze speech samples generated from latent samples drawn from the prior.

The paper is well-written and easy to follow, but the significance of the main contribution of the paper remains unclear to me. The authors propose to use an augmented standard VAE loss with a capacity hyperparameter and a Lagrange multiplier, but such modifications have been used before and it is not clear to me where is the novelty in there?

Moreover, the authors treat the Lagrange multiplier as a learnable parameter, which brings up the question how does it effect the learning dynamics. For instance if the KL-divergence reaches its capacity, the Lagrange multiplier may be pushed down to zero, which in turn can allow the posterior to diverge unchecked to possibly a point mass distribution? The authors provide no details on how beta (the Lagrange multiplier) evolves in their experience and how it effects the stochastic nature of the model.

I am not sure why the authors call non-stochastic latent variable models heuristic (instead of deterministic) methods?

In Figure 1, how are  the loss values computed for variational methods? Can we see how the error bars look for different C values and embedding dimensions?

Can the authors be more clear about why for the tasks they consider, a standard (deep) VAE architecture with non-hierarchical latents does not sufficiently capture variations in the data? Can the authors quantify the differences?

I am unfamiliar with prior work in this application area, but maybe the work is novel with respect to the application of the regularized VAE framework to speech synthesis. However, the application alone is in my opinion not a contribution that is significant enough for publication.


**Experience Assessment:**

I have read many papers in this area.

**Review Assessment: Checking Correctness Of Derivations And Theory:**

I assessed the sensibility of the derivations and theory.

**Review Assessment: Checking Correctness Of Experiments:**

I did not assess the experiments.

**Review Assessment: Thoroughness In Paper Reading:**

I read the paper at least twice and used my best judgement in assessing the paper.

---

> ### Author Response · Authors · 2019-11-12
> **Response to Review #3**
>
> Thank you for your review and your helpful comments.
>
> Our use of the Lagrange multiplier-based optimization procedure is a minor contribution of the paper as it is motivated by existing work as you point out (we mention in the paper that it's most similar to [Rezende et al., 2018] which uses a Lagrange multiplier to target a specific value for the reconstruction loss) .  We found it to be superior to existing methods for our purposes in terms of simplicity, consistency, and stability.
>
> Thank you for pointing out your concerns about the dynamics of the optimization procedure.  It is true that if the learning rate for beta was set high enough, its value could oscillate wildly or saturate, which would be undesirable.  However, we found the learning hyperparameters we chose to work consistently well for all of the capacity targets and models we tested in the paper. To help clarify the behavior of the optimization procedure, we added plots to the appendix showing how beta and the KL term evolve during training for a variety of capacity targets.
>
> We borrow the term "heuristic approaches" from [Henter et al., 2018], which is a thorough overview of related work.  In that paper, the authors use the term to refer to methods that use an unsupervised training heuristic (e.g., auto-encoder loss) but do not have an explicit prior over latent codes.  They don't have to be deterministic because the decoder can still be probabilistic.
>
> For the preliminary comparison in Figure 1, we compute the reconstruction loss term on a validation set that is the same size as the test set (either the full autoencoder loss for the heuristic methods or the negative ELBO minus the KL term for the variational approaches).  What type of error bars do you think would be useful to see here?
>
> The non-hierarchical results presented in Section 4.3 (preceding the "Hierarchical results" paragraph) were meant to show that a deep non-hierarchical VAE does adequately capture the variation in the data.  In any case, the use of hierarchical structure gives us a new ability: it allows the amount of information specified by z_H about the variation in the data to be adjusted using C_H, and the remaining variation (capped at C_L) to be sampled from the model.  This allows a flexible tradeoff between transfer fidelity and sample-to-sample variation as demonstrated in Figure 5.
>
> We think this work represents a significant contribution to the field of end-to-end speech synthesis. :)

---

### Author Response · Authors · 2019-11-12
**Uploaded a revised version of the paper**

Thank you to the reviewers for your helpful comments. We have uploaded a revised version of the paper containing the following changes:
1. Added comments about the updates to beta and the updates to the model parameters being interleaved during training to the description of eq. (9) and the "Training" paragraph of Section 4.1.
2. Added plots demonstrating the evolution of beta and the KL term for different capacity limits (Figure B.2 in the appendix) along with additional explanations in Appendix B "Optimization Examples".
3. Slight changes to the description of eqs. (10-14) that should help clarify the difference between R_* and R^{AVG}_*.
4. Modified the explanations about the layout of the latent space when using incomplete posteriors to Section 4.3.
5. Added clarification about how our approach to hierarchical latents compares to past work (5th paragraph of Section 1, 2nd paragraph of Section 3.2).

We're currently discussing how best to incorporate the remaining feedback into subsequent revisions and we encourage the reviewers to respond with additional comments if desired.

---

### Comment · Area_Chair1 · 2019-11-14
**Reviewers, any comments on the author responses?**

Dear Reviewers, thanks for your thoughtful input on this submission!  The authors have now responded to your comments.  Please be sure to go through their replies and revisions.  If you have additional feedback or questions, it would be great to get them this week while the authors still have the opportunity to respond/revise further.  Thanks!

---

### Public Comment · ~Zining_Zhang1 · 2020-05-18
**Question about the lagrange term**

we limit beta to be greater than 0, and we want to calculate the max on the loss.
If the term (KL-C) is greater than 0 as stated in equation (8), the to reach max, beta can be arbitrarily large.
Thus in this lagrange primal problem, the constraint should be (KL-C) <= 0, I suppose.
and this is contrary to the theory that the KL term is the upper bound to mutual information or capacity limit.
Thus I am wondering whether, in equation 9, the correct form should be beta(C-KL) instead of beta(KL-C)?

Please correct me if I am wrong, thanks.

---

### Decision · Program_Chairs · 2019-12-19

**Decision:**

Reject

**Comment:**

This paper investigates variational models of speech for synthesis, and in particular ways of making them more controllable for a variety of synthesis tasks (e.g. prosody transfer, style transfer).  They propose to do this via a modified VAE objective that imposes a learnable weight on the KL term, as well as using a hierarchical decomposition of latent variables.  The paper shows promising results and includes a good amount of analysis, and should be very interesting for speech synthesis researchers.  However, there is not much novelty from a machine learning perspective.  Therefore, I think the paper is not a great fit for ICLR and is better suited for a speech conference/journal.